# Osteoblast Growth in Quaternized Silicon Carbon Nitride Coatings for Dental Implants

**DOI:** 10.3390/ma17215392

**Published:** 2024-11-04

**Authors:** Haochen Zhu, Xinyi Xia, Chao-Ching Chiang, Rachael S. Watson Levings, Justin Correa, Fernanda Regina Godoy Rocha, Steve C. Ghivizzani, Fan Ren, Dan Neal, Patricia dos Santos Calderon, Josephine F. Esquivel-Upshaw

**Affiliations:** 1Department of Chemical Engineering, College of Engineering, University of Florida, Gainesville, FL 32611, USA; zhu.haochen@ufl.edu (H.Z.); cchiang@ufl.edu (C.-C.C.); fren@che.ufl.edu (F.R.); 2Department of Molecular Genetics & Microbiology, College of Medicine, University of Florida, Gainesville, FL 32610, USA; watsors@ortho.ufl.edu (R.S.W.L.); ghivisc@ortho.ufl.edu (S.C.G.); 3Department of Restorative Dental Sciences, Division of Prosthodontics, College of Dentistry, University of Florida, Gainesville, FL 32610, USA; 4Department of Periodontology, College of Dentistry, University of Florida, Gainesville, FL 32610, USA; fgodoyrocha@dental.ufl.edu; 5Department of Surgery, College of Medicine, University of Florida, Gainesville, FL 32610, USA; dneal@ufl.edu; 6Department of Dentistry, Federal University of Rio Grande do Norte, Natal 59056, RN, Brazil; patriciascalderon@yahoo.com.br

**Keywords:** nanotechnology, dental implants, surface coatings, bacterial adhesion, antimicrobial, biocompatibility

## Abstract

The demand for dental implants has increased, establishing them as the standard of care for replacing missing teeth. Several factors contribute to the success or failure of an implant post-placement. Modifications to implant surfaces can enhance the biological interactions between bone cells and the implant, promoting better outcomes. Surface coatings have been developed to electrochemically alter implant surfaces, aiming to reduce healing time, enhance bone growth, and prevent bacterial adhesion. Quaternized silicon carbon nitride (QSiCN) is a novel material with unique electrochemical and biological properties. This study aimed to assess the influence of QSiCN, silicon carbide nitride (SiCN), and silicon carbide (SiC) coatings on the viability of osteoblast cells on nanostructured titanium surfaces. The experiment utilized thirty-two titanium sheets with anodized TiO_2_ nanotubes featuring nanotube diameters of 50 nm and 150 nm. These sheets were divided into eight groups (n = 4): QSiCN-coated 50 nm, QSiCN-coated 150 nm, SiCN-coated 50 nm, SiCN-coated 150 nm, SiC-coated 50 nm, SiC-coated 150 nm, non-coated 50 nm, and non-coated 150 nm. Preosteoblast MC3T3-E1 Subclone 4 cells (ATCC, USA) were used to evaluate osteoblast viability. After three days of cell growth, samples were assessed using scanning electron microscopy (SEM). The results indicated that QSiCN coatings significantly increased osteoblast proliferation (*p* < 0.005) compared to other groups. The enhanced cell adhesion observed with QSiCN coatings is likely due to the positive surface charge imparted by N^+^.

## 1. Introduction

Dental implants are an important tool for the replacement of teeth in modern dentistry. Complete or partial edentulism is a public health concern, with 26% of the population expected to lose all their teeth after the age of 74 [1]. There are multiple advantages that implants have over fixed dental prostheses, which include high success rates, decreased risks of dental diseases and sensitivity to adjacent teeth, and bone maintenance in edentulous areas [2]. There is a growing interest in being able to restore masticatory functions and esthetic appearances without having to manipulate healthy teeth or take in and out a removable appliance. Researchers continue to improve the designs and materials used in dental implants to improve the efficiency and predictability of these restorations.

The fundamental concept underlying dental implants is osseointegration, defined as the direct interaction between bone and the implant surface. This interaction initiates a series of biological processes, including angiogenesis, extracellular matrix deposition, and the recruitment of osteogenic cells, all of which promote bone growth along the implant surface [3]. Titanium (Ti) and zirconia (Zr) are widely used implant materials due to their excellent biocompatibility and strong capacity for osseointegration within the alveolar bone. Continuous advancements are being made to accelerate the osseointegration process, as the speed at which an implant integrates with bone is critical to its long-term clinical success. Faster osseointegration not only enhances the stability of the implant but also allows for quicker functional restoration, offering patients immediate improvements in oral health and functionality following implant placement.

Anodization is a surface modification technique used to enhance the surface area of dental implants, promoting osteoblast growth [4] and improving osseointegration. During this process, titanium implants are immersed in an aqueous electrolytic solution and subjected to an applied voltage, which forms titanium oxide (TiO_2_) nanopores or nanotubes on the surface [5]. The increased surface roughness and surface area created by these nanotubes enhance cellular activity. Khaw et al. [6] investigated the relationship between TiO_2_ nanotube diameter and osteoblast proliferation, comparing nanotubes with diameters of 20 nm, 50 nm, and 100 nm. Their findings revealed that larger nanotubes (50 nm and 100 nm) supported higher metabolic activity and cell growth. Similarly, Bandyopadhyay et al. [7] demonstrated how TiO_2_ nanotubes with a diameter of 100 nm improved osteoblast attachment, proliferation, and differentiation.

The impact of surface coatings further enhances cellular activity, but the size of the nanotubes remains a crucial factor in promoting osteoblast function. Lee et al. [4] fabricated TiO_2_ nanotube arrays on titanium dental implants using anodic oxidation and incorporated recombinant human bone morphogenetic protein-2 (rhBMP-2) into some nanotubes. Implants loaded with rhBMP-2 exhibited the highest bone-to-implant contact ratio, while even those with only nanotube arrays outperformed traditional machined surface implants. This suggests that both the size of the nanotubes and the presence of surface coatings contribute significantly to enhancing osteoblast activity and implant performance.

According to a study from MS Howe, the estimated 10-year survival rate for dental implants was 96.4% [8]. Two of the major reasons for the failure of dental implants are chemical corrosion and bacterial infection. A substantial coating that is anti-corrosive and antibacterial can be an effective way to prevent these problems.

Silicon carbide (SiC) is valued for its mechanical strength, stability, and biocompatibility. Hsu [9] demonstrated that SiC coatings on TiO_2_ nanotubes (ATO) improved corrosion resistance in a 3.5% NaCl solution compared to non-coated surfaces. Calderon [10] found that SiC coatings enhanced osteoblast cell proliferation and mineralization on anodized titanium surfaces while also offering notable antibacterial properties [11].

Silicon carbon nitride (SiCN), produced by incorporating nitrogen into SiC, is a versatile material with strong antibacterial properties. By adjusting the ratios of silicon, carbon, and nitrogen, SiCN can be tailored for various applications. Its hydrophobic surface can inhibit bacterial adhesion [12,13], as demonstrated by Petersson et al. [14], who applied SiCN coatings to artificial joints, improving both biocompatibility and longevity.

Quaternization, the addition of an N+ ion, further enhances the antibacterial properties of materials by disrupting negatively charged bacterial cell membranes [15]. Quaternized SiCN (QSiCN) is formed by adding quaternary nitrogen compounds to SiCN (R_3_N + RCl → R_4_N^+^Cl^−^). QSiCN’s positive charge provides potent electrochemical properties that destroy bacteria on surfaces. Carey et al. [16] showed that quaternized titanium nitride (QTiN) coatings reduced bacterial growth compared to TiN- and Ti-coated surfaces, while Camargo et al. [17] found QSiC to be even more effective. Chiang et al. [18] further enhanced QSiCN’s antibacterial effects by adjusting the degree of nitrogen quaternization, with electron microscopy confirming bacterial cell membrane destruction as the mechanism of action.

For an antibacterial and anti-corrosive coating to be successful, this coating material should also promote bone growth and not prove toxic to bone-forming cells. In this study, samples with TiO_2_ nanotubes were coated with either (1) QSiCN; (2) SiCN; or (3) SiC. Contact angle measurements and cell viability tests were performed with osteoblast cells to determine whether these coatings could not only prevent bacterial infection but also maintain the biocompatibility of TiO_2_ nanotubes.

## 2. Materials and Methods

### 2.1. Experimental Design

Thirty-two titanium sheets with anodized TiO_2_ nanotubes (InRedox, Longmont, CO, USA) and two smooth-surfaced titanium disks (MilliporeSigma, Burlington, MA, USA) were used in this study. The nanotubes had diameters of 50 nm and 150 nm, as shown in Figure 1. Eight experimental groups (n = 4) were created, utilizing coatings based on quaternized silicon carbon nitride (50-TiO_2_-QSiCN and 150-TiO_2_-QSiCN), silicon carbon nitride (50-TiO_2_-SiCN and 150-TiO_2_-SiCN), silicon carbide (50-TiO_2_-SiC and 150-TiO_2_-SiC), and uncoated surfaces (50-TiO_2_ and 150-TiO_2_). The applied coatings were approximately 10 nm thick on each side, reducing the nanotube diameters by 20 nm.

### 2.2. Coating and Quaternization Process

Plasma-enhanced chemical vapor deposition (PECVD) was used to coat the titanium nanotube sheets with SiC or SiCN. The gas precursors applied to the SiC/SiCN deposition and their respective deposition rates were methane (CH_4_, 100 sccm), silane (SiH_4_, 300 sccm), helium (He, 700 sccm), and ammonia. (NH_3_, 2 sccm, only used in SiCN deposition). The deposition temperature and bias voltage was 350 °C and 15 V, respectively. Once the coating process was completed, the samples were cut into 1 cm squares using scissors and cleaned with acetone (Thermo Fisher Scientific, Pittsburgh, PA, USA) and isopropyl alcohol (IPA, Thermo Fisher Scientific, Pittsburgh, PA, USA) to remove surface contamination. The titanium nanotubes with a SiCN coating were then immersed in a mixture of acetonitrile (25 mL, Thermo Fisher Scientific, Pittsburgh, PA, USA) and allyl bromide (100 µL, Thermo Fisher Scientific, Pittsburgh, PA, USA) on a belly dancer (Stovall, Greensboro, NC, USA) for an hour to make the nitrogen atoms on the surface of the SiCN coating turn into quaternary nitrogen.

### 2.3. Cell Viability

Preosteoblast MC3T3-E1 Subclone 4 cells (ATCC CRL-2593, Washington, DC, NW, USA) were cultured in an incubator with 5% CO_2_ at 37 °C. The base medium for the cells was 1× Alpha Minimum Essential Medium with ribonucleosides, deoxyribonucleosides, 2 mM L-glutamine, and 1 mM sodium pyruvate, but without ascorbic acid (MEM Alpha, Thermo Fisher Scientific, Pittsburgh, PA, USA). In total, 10% of fetal bovine serum (Thermo Fisher Scientific, Pittsburgh, PA, USA) and 1% of penicillin (Thermo Fisher Scientific, Pittsburgh, PA, USA) and streptomycin (Thermo Fisher Scientific, Pittsburgh, PA, USA) were added to the medium to enable cell growth. All cells used in the experiment were at passages lower than 15. Four samples of each experimental group and two titanium disks as control groups were placed on a sterile 24-well plate and were autoclaved for 30 min. Cells with the concentration of 2 × 10^4^ cells/mL were added into each well and cultured on the samples for 3 days.

Cell viability was measured by CyQUANT™ MTT Cell Viability Assay (V-13154, Thermo Fisher Scientific, Pittsburgh, PA, USA), which was performed under the instructions of the manufacturer using a Spectronic 20D+ Spectrophotometer (Thermo Fisher Scientific, Pittsburgh, PA, USA). After 3 days of growth, cells were detached from the surface by a 1 mL trypsin-EDTA (Thermo Fisher Scientific, Pittsburgh, PA, USA) solution for 1 min at 37 °C, and then trypsin was deactivated by adding 0.5 mL of a fresh α-MEM medium solution.

### 2.4. Scanning Electron Microscopy

Titanium nanotube sheets were inspected by scanning electron microscopy (FEI NOVA NanoSEM 430, FEI Company, Hillsboro, OR, USA) with a secondary electron through-the-lens detector (TLD) to confirm the cell attachment on the surface. The acceleration voltage was 5 kV, and the samples were sputtered with 10 nm of gold before the observation. One sample from each experimental group was used for the SEM observation after the 3-day cell growth.

The osteoblast cells that attached to the surface were fixed in the solution of 3% glutaraldehyde (50 wt.% in DI water), 0.1 M sodium cacodylate, and 0.1 M sucrose for 45 min and then immersed in a buffer solution of 0.1 M sucrose and 0.1 M sodium cacodylate for 10 min. Samples were immersed in ethanol with a serial concentration (30%, 50%, 70%, and 100%) for 10 min each, and hexamethyldisilane (HMDS) was utilized for the dehydration of the samples. Finally, the samples were coated with a 10 nm palladium–gold alloy to prevent the charging of the surface.

### 2.5. Data Analysis

Quantitative data are shown as the means ± standard deviations. Statistical analysis was performed using XLSTAT free version for Windows (Lumivero Inc., Denver, CO, USA) (https://www.xlstat.com/en/download, access date 22 January 2024). Statistical differences were calculated using the Kruskal–Wallis and Mann–Whitney tests. A *p*-value of 0.05 was considered statistically significant.

## 3. Results

### 3.1. Cell Viability

The cell viability of osteoblast cells was measured by the MTT test. Absorbance at a 562 nm wavelength was obtained for all the eight experimental groups, and one reference group (Ti disk without nanotubes) (Table 1). The results demonstrate that cell viability is significantly higher in samples with QSiCN coatings (*p* < 0.0001). This indicates that QSiCN coatings enhance the adhesion between osteoblast cells and Ti. The paired significance of QSiCN with other coatings was as follows: (1) QSiCN vs. SiCN (*p* = 0.002); (2) QSiCN vs. SiC (*p* = 0.003); and (3) QSiCN vs. Ti (*p* = 0.0006). The 50 nm size nanotubes had better cell viability than 150 nm size nanotubes except for SiC, although differences between 50 nm and 150 nm size nanotubes were negligible for all coatings except QSiCN.

### 3.2. Cell Attachment

Figure 2 demonstrates the osteoblast attached to the sample surface after 3 days of cell proliferation at 800× magnification. The cells were oval-shaped and flattened on the sample surface, and there was no significant difference in the cell coverage rate between different samples, which was consistent with the MTT results.

Figure 3a illustrates more details of a single osteoblast attached to the QSiCN sample surface, and the interaction between the cell and TiO_2_ nanotubes can be seen in Figure 3b.

## 4. Discussion

In this article, QSiCN-coated TiO_2_ nanotubes were tested to evaluate their biocompatibility to osteoblast compared to different kinds of coating materials. Commonly used methods to improve the performance of commercial dental implants [19] are shown in Table 2. These methods can increase the surface roughness of dental implants to facilitate osseointegration. However, none of these methods can provide antibacterial properties, and many of them have poor performance over long-term use [20,21]. As shown in Table 1, anodization alone is insufficient to establish an environment that could inhibit the colonization of bacterial species and prevent the deterioration of implant materials. Additional methods, such as applying surface coatings to implants, are imperative to enhance the biochemical and mechanical properties that promote clinical success.

The QSiCN-coated TiO_2_ nanotubes tested in our study significantly increased osteoblast proliferation. According to a previous study conducted by Chiang et al. [18], the contact angle of QSiCN was about 90°, which is 15° larger than SiCN. This incremental increase in hydrophobicity can be explained by the extension of the allyl group on the sample surface. Changing the nitrogen content of the QSiCN coating does not affect the surface wettability [18]. Its highly hydrophobic surface is less favorable for bacteria to adhere to, which is the reason why QSiCN has remarkable antibacterial properties.

The positively charged nitrogen atoms on the surfaces of QSiCN-coated TiO_2_ nanotubules also play an important role in facilitating antibacterial activity. These atoms can oxidize the lipids in bacterial cell membranes and cause lipid peroxidation [22]. Lipid peroxidation is a biochemical process in which free radical oxidants attack lipids that contain carbon–carbon double bonds, which ultimately disrupts the membrane permeability barrier function [23,24]. As the QSiCN-coated TiO_2_ nanotubules interact with bacterial cell membranes, the permeability of the membrane is increased by lipid peroxidation, and its constituents are then released from the cytosol, leading to cell death (Figure 4).

The differences in cellular membrane composition between bacterial cells and human osteoblasts are significant, particularly in the presence of cholesterol. Eukaryotic cell membranes, such as those of osteoblasts, contain higher amounts of cholesterol—approximately 30% of the lipid bilayer. This cholesterol is critical for maintaining membrane rigidity and fluidity, which is essential for the proper functioning of membrane proteins involved in physiological processes [25,26]. Additionally, cholesterol plays a key role in preventing lipid peroxidation by decreasing membrane fluidity and promoting the formation of lipid rafts, which are important for signal transduction and cell stability [27].

The positively charged surface of QSiCN-coated TiO_2_ implants interacts favorably with osteoblast cells, whose membranes carry a net negative charge. This electrostatic attraction enhances the adhesion of osteoblasts to the implant surface without causing damage to the cell membrane. Rather than being harmful, the positive charges on QSiCN coatings help facilitate this interaction, promoting osteoblast attachment and proliferation. This property likely explains why QSiCN-coated surfaces perform better in MTT assays, which measure cell viability compared to other coating materials [28]. By fostering better cell adhesion and viability through electrostatic interactions, QSiCN coatings show promise for enhancing the integration of implants into bone tissue (Figure 5).

Biocompatibility is an important property of dental implants [29]. Both MTT data and SEM images indicate that QSiCN surface coatings facilitate cell adhesion to substrate material. Previous studies on SiC-coated titanium implants [30,31,32,33,34] and SiCN-coated titanium implants [35] all confirm that these coating materials have good biocompatibility to osteoblast cells. In this study, QSiCN had better performance than SiC and SiCN in these aspects. Future research should focus on further optimizing the electrostatic interactions between QSiCN coatings and osteoblast cells by adjusting the degree of quaternization or altering surface roughness and nanotube dimensions. Long-term in vivo studies are also needed to assess how QSiCN coatings affect not only initial cell adhesion but also bone remodeling, osseointegration durability, and resistance to bacterial infection over time.

## 5. Conclusions

The use of dental implants to replace missing teeth has become a widely accepted and effective treatment method. Ongoing research aims to further optimize implants for enhanced clinical success and longevity by improving the biological processes that ensure a stable connection between implant surfaces and host tissues. This study demonstrates that all coating materials—SiC, SiCN, and QSiCN—promote osteoblast adhesion on TiO_2_ nanotubular surfaces, with each showing favorable biocompatibility. While QSiCN-coated surfaces exhibit the strongest cell adhesion and proliferation, both SiC and SiCN also contribute positively to cell attachment and growth. SEM images provide further evidence of these cells interacting with the coatings, highlighting their potential for enhancing osseointegration. Interestingly, the size of the TiO_2_ nanotubes did not significantly affect cell viability, suggesting that the surface coating, rather than the nanotube diameter, may play a more decisive role in promoting osteoblast activity.

## Figures and Tables

**Figure 1 materials-17-05392-f001:**
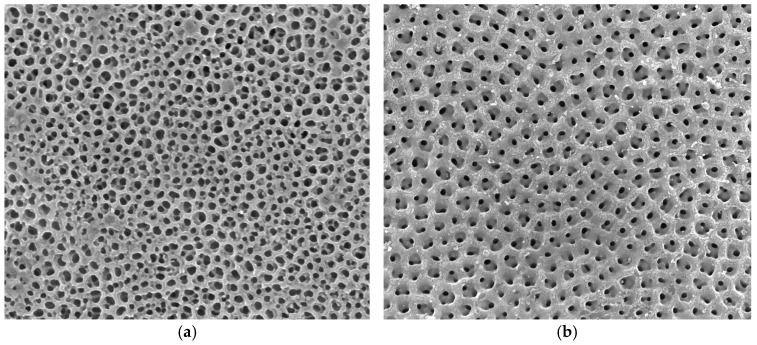
SEM images showing nanotubes on the sample surface coated with QSiCN. (**a**) 50 nm; (**b**) 150 nm. In total, 10 nm of the coating was applied to either side of the tube for a total of 20 nm.

**Figure 2 materials-17-05392-f002:**
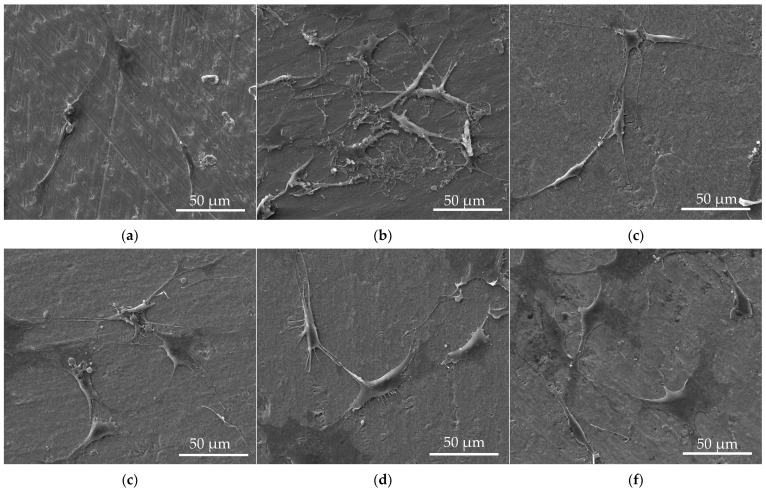
SEM images of osteoblast cells on sample surface after 3 days of culture. (**a**) Ti; (**b**) 50 nm-TiO_2_; (**c**) 150 nm-TiO_2_; (**d**) 50 nm-SiC; (**e**) 150 nm-SiC; (**f**) 50 nm-SiCN; (**g**) 150 nm-SiCN; (**h**) 50 nm-QSiCN; and (**i**) 150 nm QSiCN.

**Figure 3 materials-17-05392-f003:**
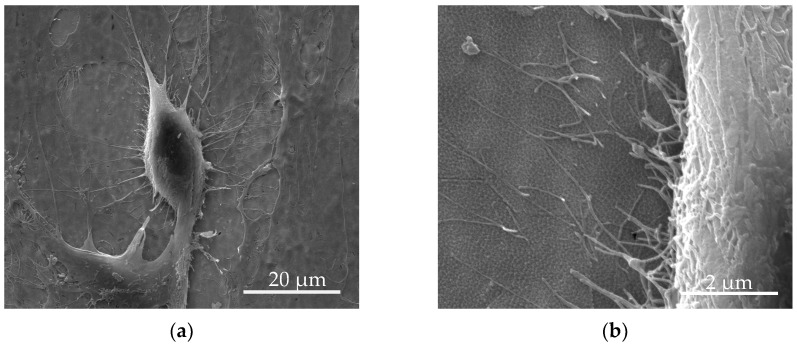
SEM images of single osteoblast on QSiCN samples: (**a**) 2500× magnification; (**b**) 20,000× magnification.

**Figure 4 materials-17-05392-f004:**
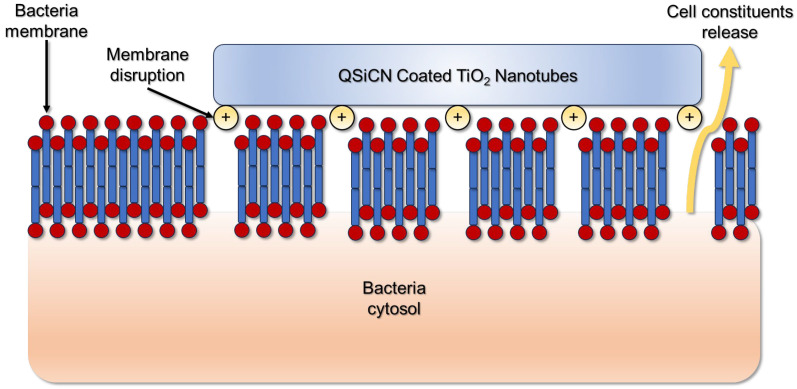
Schematic of the mechanism of how the QSiCN surface kills bacteria.

**Figure 5 materials-17-05392-f005:**
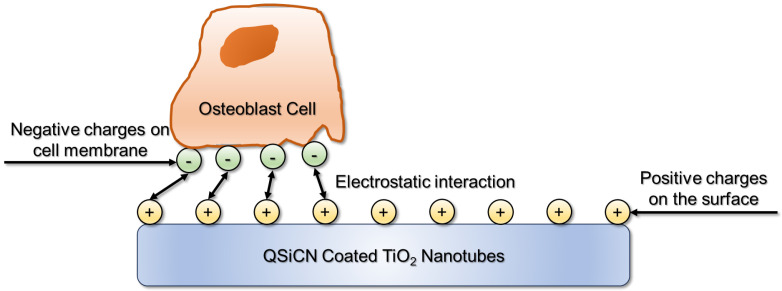
Schematic showing the interaction between osteoblasts and QSiCN-coated nanotubes.

**Table 1 materials-17-05392-t001:** Cell growth (mean ± SD).

Group	50 nm	150 nm
Non-coated	0.079 (±0.012)	0.074 (±0.007)
SiC-coated	0.077 (±0.007)	0.083 (±0.008)
SicN-coated	0.086 (±0.014)	0.079 (±0.006)
QSiCN-coated	0.103 (±0.017)	0.086 (±0.007)

**Table 2 materials-17-05392-t002:** Commonly used methods in commercial dental implants.

Surface Modifications	Indication	Currently Being Used	Osseointegrative	Anti-Bacterial	Anti-Corrosive	Disadvantages
Sand blasting	Improves topography and wettability	Most currently commercialized method together with acid etching	Microspheres of TiO_2_, Al_2_O_3_, SiO_2_, and HA increase surface roughness and osseointegration	No	No	20-year follow- did not improve bone healing but increased peri-implantitis and failure
Acid Etching	Improves topography and wettability	Together with sandblasting, produces SLA surface	Micro-pits surface with immersion in corrosive acids such as HCl, H_2_SO_4_, HNO_3_, and HF	No	No	
Anodization	Improves topography and wettability	TiUnite^®^ (Nobel Biocare, Gothenburg, Sweden)	Oxidation of Ti surface to increase surface area	No	No	Tribocorrosion has been observed
Plasma Spraying	Improves biocompatibility and protein absorption	ITI Straumann	Additional layer of OH groups increase osseointegration	No	No	HA debonds with time
Laser Ablation		Laser-Lok^®^ (BioHorizons, Birmingham, AL, USA). SLActive^®^ (Straumann Institute, Basel, Switzerland)		No	No	

## Data Availability

The original contributions presented in the study are included in the article, further inquiries can be directed to the corresponding author.

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
