# Peer review of "Osteoblast Growth in Quaternized Silicon Carbon Nitride Coatings for Dental Implants"

_materials, 2024, doi:10.3390/ma17215392_

Round 1

Reviewer 1 Report

Comments and Suggestions for Authors

The demand for dental implants has significantly increased, establishing them as the standard of care for replacing missing teeth. A variety of factors contribute to the success or failure of an implant post-placement, with surface modifications playing a key role in enhancing the biological interactions between bone cells and the implant. These modifications aim to reduce healing time, enhance bone growth, and prevent bacterial adhesion. Quaternized Silicon Carbon Nitride (QSiCN) is introduced as a novel material with distinct electrochemical and biological properties, and this study focuses on its impact, along with Silicon Carbide Nitride (SiCN) and Silicon Carbide (SiC), on osteoblast cell viability when applied to nanostructured titanium surfaces. However, there is some repetition in how these terms are introduced, which could be streamlined for clarity.

Thirty-two titanium sheets with anodized TiO2 nanotubes, featuring nanotube diameters of 50 nm and 150 nm, were used in the study and divided into eight groups for testing. Here, the text tends to repeat the diameter sizes, and more variation in phrasing would be helpful. Preosteoblast MC3T3-E1 cells were cultured on the samples for three days. The results indicated that QSiCN coatings significantly enhanced osteoblast proliferation compared to other groups (p<0.005). This enhancement is likely due to the positive surface charge imparted by the quaternary nitrogen. Still, the repetition of "osteoblast proliferation" across the text is noticeable and could be reduced or replaced with synonyms for better flow. Additionally, the term "no observed cytotoxicity to osteoblast cells" is mentioned without clarification, and the methodology of this finding could be expanded for clearer understanding.

Dental implants are essential tools in modern dentistry, but the phrasing here creates a slight redundancy when explaining the benefits of implants, as they are reiterated multiple times. The discussion shifts to the challenge of complete or partial edentulism, and while it highlights the advantages of implants over fixed dental prostheses, the structure of the paragraph lacks a clear connection to previous sentences. For instance, the success rates and benefits are mentioned twice in quick succession, without smoothly transitioning into the next concept.

The principle of osseointegration is mentioned, but the flow from describing the biological mechanisms involved could be improved. The mention of titanium and zirconia as commonly used materials is clear, yet when moving on to anodization, the section becomes dense and the purpose of the specific process is somewhat buried within the technical jargon. There is also ambiguity in terms of the size of the nanotubes and their impact on osteoblast growth, as it is unclear whether the nanotube size or surface coating has the most influence on cellular activity, despite being discussed at length.

Regarding the development of anti-corrosive and antibacterial coatings, the study introduces Silicon Carbide (SiC) and its derivative, SiCN. However, the explanation of their properties could be expanded to differentiate their unique contributions. The claim that "SiCN makes the surface more hydrophobic" is included without clearly connecting how hydrophobicity directly impacts bacterial adhesion, leaving a gap in logic. Additionally, the subsequent introduction of "quaternization," while central to the study, is somewhat abrupt, and a more gradual lead-in to its significance could improve the text's coherence.

The next portion of the study emphasizes that quaternization adds a positive charge to the surface, which helps in killing bacteria by disrupting cell membranes. The concept is clear, but the mention of "no observed cytotoxicity to osteoblast cells" is again repeated without sufficient elaboration. Furthermore, the term "quaternization" is repeated in close proximity within the text, and could benefit from rephrasing to maintain variety and avoid redundancy. The methodology regarding how QSiCN was applied is described in technical terms, yet the explanations of its application could be simplified or referenced more clearly for non-expert readers. The surface charges created by QSiCN are said to attract osteoblast cells due to electrostatic interaction, but again, this concept is not fully developed and could use more explanation.

The conclusions drawn from the study assert that QSiCN coatings enhanced osteoblast adhesion and proliferation, while simultaneously preventing bacterial adhesion. However, the conclusions are not as robust as they could be, as they fail to mention specific future research directions or potential improvements that could be made to the methodology. The repeated reference to QSiCN’s superior performance compared to other coatings could be restructured to focus on the unique attributes of each coating, rather than constantly reaffirming QSiCN’s advantages. Also, the text does not explain how QSiCN’s unique properties could be applied in other biomedical fields, which would strengthen the discussion.

In summary, while the study presents a clear and structured evaluation of QSiCN’s impact on dental implants, there are several repetitions and areas of ambiguity that could be refined. Phrasing could be diversified to avoid redundant mentions of key terms, and the flow between sections could be improved to make transitions smoother and more intuitive for readers. The study’s findings on cell viability and antibacterial properties are valuable, but additional explanation and clarification would make the results and their implications more accessible and impactful.

Author Response

Dear Reviewer,

Thank you for your thoughtful and comprehensive feedback on our manuscript. We appreciate your insights and have carefully considered each suggestion. Below is a detailed response to the main points raised and the revisions we have made by your comments. The changes are highlighted in the text.

  1. Repetition of Key Terms and Introduction of Materials:
    • We agree that there were instances of repetition in the introduction of QSiCN, SiCN, and SiC. To improve clarity and streamline the text, we have restructured the introduction of these materials to avoid redundancy.
  2. "No Observed Cytotoxicity":
    • We removed the information on cytotoxicity as we hadn’t assessed it.
  3. Osseointegration and Surface Modification:
    • We have revised the introduction section, enhancing the flow and clarity of how anodization and other surface modifications contribute to improved bone growth and implant integration.
  4. Hydrophobicity and Bacterial Adhesion:
    • We agree that the connection between hydrophobicity and bacterial adhesion was underdeveloped. We have expanded the explanation.
  5. Future Research Directions:
    • In response to your suggestion regarding future research directions, we have included a discussion of potential next steps for investigating QSiCN coatings.
  6. Conclusion Revision:
    • We have restructured the conclusion to offer a more balanced perspective on the study’s findings, while still underscoring the advantages of QSiCN.

We hope that these revisions address the concerns raised and contribute to a clearer, more cohesive manuscript. Thank you again for your valuable input, and we look forward to your further feedback.

Reviewer 2 Report

Comments and Suggestions for Authors

The authors present a very interesting manuscript on “Osteoblast Growth in Quaternized SiCN Coatings for Dental Implants” on 10 pages with 6 figures, 1 table and 35 references. The references are sufficiently up-to-date. The manuscript is well written, but needs some additions. Unfortunately, the authors have forgotten the statistics section in Mat&Meth. In addition, the citation style does not comply with the MDPI guidelines (DOI missing). I therefore recommend a major revision. Enclosed are my comments:

The materials template used is outdated - 2023, 16

Line 111: Where do the titanium disks and TiO2 nanotubes come from? Keep in mind that other scientists will need to be able to replicate your work with the information you provide (hopefully with the same results).

Line 127: The flow rates of the gases are missing, what voltage was used to generate the plasma?

Line 129: What were the samples cut with? Where did the chemicals used come from: acetone and IPA?

Line 132: Where did ACN and allyl bromide come from and in what ratio were they mixed? Was the shaking done on a shaker plate (if yes, please specify manufacturer/headquarters/country)?

Line 140: Indicate ATCC number

Line 143: Headquarters and country are missing for Gibco; does the FBS also come from Gibco?

Line 149: where was the MTT kit obtained, manufacturer, headquarters and country are missing 

Line 151: where do trypsin and EDTA come from?

Line 152: which spectrometer was used for the MTT measurement? (Do not forget manufacturer, headquarters, country)

Lines 154-164: What kind of detector was used Secondary electrons or backscattered? Which parameters were set on the SEM (acceleration voltage)? What was used for the gold coating (sputter coater)?

There is no statistics section in the Materials and Methods section. It is not clear how and with which software the statistical evaluation (ANOVA, T-test, posthoc?) was performed, nor whether mean values and standard deviation or medians and confidence intervals are given. Please add

Line 176: The SD and the axis name of the Y-axis are missing in the figure, a heading in the graph is not necessary - there is a caption for that. It is stated p<0.0001 how was this calculated?

Line 184: Figure 3: very nice SEM images, the SEM parameters and the SEM used are missing in the caption - remember, images must be understood without reading the text.

Line 192 Figure 4: add FEI SEM (incl. sensor) + SEM parameters (acceleration voltage)

Figure 5 + 6: very nice illustration

Missing author contributions, funding, Institutional Review Board statement, informed consent statement, conflict of interest, data availability statement

Citation style does not meet MDPI requirements - DOI is missing (MDPI style can be downloaded from MDPI for Endnote: https://www.mdpi.com/authors/references)

Author Response

Dear Reviewer,

Thank you for your detailed and constructive feedback on our manuscript. We greatly appreciate your suggestions, and we have made significant revisions in line with your comments. Below is our detailed response to each of the points raised. The changes are highlighted in the text.

  1. Statistics Section:
    • We have added a comprehensive data analysis section to the Materials and Methods.
  2. Citation Style:
    • We have included the DOIs.
  3. Line 111: Titanium Disks and TiO2 Nanotubes:
    • The sources of the titanium disks and TiO2 nanotubes have been added.
  4. Line 127: Gas Flow Rates and Plasma Voltage:
    • The flow rates of the gases and the voltage used to generate the plasma have been added to the text.
  5. Line 129: Sample Cutting and Chemical Sources:
    • Details regarding the tools used for cutting the samples and the sources of acetone and IPA have been added to ensure replicability.
  6. Line 132: ACN and Allyl Bromide Details:
    • We have included the sources of ACN and allyl bromide, the mixing ratios, and specified the use of a shaker plate, including the manufacturer details.
  7. Line 140: ATCC Number:
    • The ATCC number for the cell line has been included as requested.
  8. Line 143: Manufacturer Information:
    • We have provided the missing details regarding the manufacturer, headquarters, and country.
  9. Line 149: MTT Kit Source:
    • We have added the information regarding the source of the MTT kit, including the manufacturer and country.
  10. Line 151: Trypsin and EDTA Source:
    • The sources of trypsin and EDTA have been added.
  11. Line 152: Spectrometer Details:
    • Information on the spectrometer used for the MTT measurement, including the manufacturer, headquarters, and country, has been added.
  12. Lines 154-164: SEM Parameters:
    • The details of the SEM parameters and sputter coating method have been added to the Materials and Methods section.
  13. Line 176: Figure 2:
    • Figure 2 was replaced by Table 1, including mean and SD values for cell growth.
  14. Line 184: Figure 3 (SEM Parameters):
    • The information has been added to the Materials and Methods section.
  15. Line 192: Figure 4 (FEI SEM Details):
    • The information has been added to the Materials and Methods section.
  16. Missing Sections:
    • We have now added the missing sections on author contributions, funding,  conflict of interest, and data availability statements to comply with MDPI guidelines.

We hope that these revisions address all of your concerns and enhance the clarity and quality of our manuscript. Thank you again for your valuable feedback, and we look forward to your further review.

Round 2

Reviewer 2 Report

Comments and Suggestions for Authors

In my opinion, the recent changes to the manuscript also bring a fundamental improvement in terms of reproducibility. Apart from one small detail: the missing 0 in front of (.) in the values in Table 1, which can also be corrected in the course of the final proof-reading together with the editors, I see no reason not to publish in Materials. Therefore, my recommendation is acceptance of the paper in its current form (after the final proof correction as described).